# Mass Spectrometry-Based Structural Proteomics for Metal Ion/Protein Binding Studies

**DOI:** 10.3390/biom12010135

**Published:** 2022-01-15

**Authors:** Yanchun Lin, Michael L. Gross

**Affiliations:** Department of Chemistry, Washington University in St. Louis, St. Louis, MO 63130, USA; yanchun.lin@wustl.edu

**Keywords:** mass spectrometry-based structural proteomics, HDX, FPOP, targeted amino-acid labeling, native MS, metal ion/protein interaction, binding site, binding affinity, stoichiometry, conformational change

## Abstract

Metal ions are critical for the biological and physiological functions of many proteins. Mass spectrometry (MS)-based structural proteomics is an ever-growing field that has been adopted to study protein and metal ion interactions. Native MS offers information on metal binding and its stoichiometry. Footprinting approaches coupled with MS, including hydrogen/deuterium exchange (HDX), “fast photochemical oxidation of proteins” (FPOP) and targeted amino-acid labeling, identify binding sites and regions undergoing conformational changes. MS-based titration methods, including “protein–ligand interactions by mass spectrometry, titration and HD exchange” (PLIMSTEX) and “ligand titration, fast photochemical oxidation of proteins and mass spectrometry” (LITPOMS), afford binding stoichiometry, binding affinity, and binding order. These MS-based structural proteomics approaches, their applications to answer questions regarding metal ion protein interactions, their limitations, and recent and potential improvements are discussed here. This review serves as a demonstration of the capabilities of these tools and as an introduction to wider applications to solve other questions.

## 1. Introduction

Metal ions play an essential role in biological and physiological processes, including respiration, signal transduction, and transcription [1,2]. Proteins are the basic units to carry out these processes. Approximately one-third of proteins interact with metal ions to exercise their functions [2]. Thus, understanding the molecular basis of interactions between metal ions and proteins can provide insights for these biological processes.

Biochemical and biophysical tools afford different types of binding information. Spectroscopy, fluorescence, and thermodynamic measurements can reveal binding affinity and stoichiometry [3,4,5,6]. Low-resolution protein structural characterization is often carried out by circular dichroism (CD) [7] and other spectroscopic methods. X-ray crystallography, NMR, and cryo-EM provide high-resolution structural information [8,9,10]. MS-based structural proteomics tools not only bridge the gap between low- and high-resolution structural characterization but also offer binding affinity and stoichiometry.

Protein footprinting is an important element of MS-based structural proteomics in which the solvent-exposed area of a protein is labeled in a chemical reaction [11,12]. Based on the chemical properties of the reaction, footprinting can be divided into reversible (e.g., HDX) and irreversible labeling (e.g., fast photochemical oxidation of proteins (FPOP) and other specific amino-acid covalent labeling). In addition to footprinting, other MS-based approaches including native MS and ion mobility (IM) are also commonly employed. Compared to traditional biophysical tools, MS-based tools are less demanding regarding protein size, purity, and amount, although they require expensive instrumentation and skilled operation.

In this review, we discuss the principles of MS-based structural proteomics tools; then, we illustrate their capability to characterize metal–protein interactions by discussing examples principally from our own work and some from other groups. These applications have been organized according to the different questions, in which biochemists and biophysicists might be interested, and the corresponding MS tools that can be used to answer these questions. Herein, we describe the general workflow and data-driven conclusions for each approach to enable wider application to characterize other metal ion and protein interactions. Limitations of and principles for choosing each technique are also discussed.

## 2. MS Tools for Structural Proteomics and Metal Ion Binding

Over the past two decades, a full set of MS-based structural proteomics tools has evolved to study higher-order protein structures and their changes. This section serves as an introduction to these methods. Our intention is not to provide an exhaustive review of these methods but rather to discuss the basic concepts and principles behind them as introduction to some applications.

Footprinting tools share a common feature—the solvent-exposed area of a protein is labeled with a chemical reagent and the location of the labeling is sequentially determined by bottom-up analyses (proteolysis of protein before MS analysis [13]). Based on the protein’s properties, these methods can be further divided into three major categories, that is, reversible labeling, fast irreversible radical labeling, and targeted chemical covalent labeling.

### 2.1. Hydrogen/Deuterium Exchange MS (HDX-MS)

HDX is an example of reversible labeling. Here, the protein is studied in D_2_O buffer, allowing solvent-accessible backbone amide hydrogens to exchange with deuterium [14]. Upon quench of the HDX (by bringing the solution’s pH to ~2.5), the rate reaches its minimum; thus, the deuterium is largely preserved during the analysis to follow [15]. The labeled sample can be submitted to either global protein analyses or bottom-up analyses. To acquire regional information, the labeled protein sample is denatured and then digested by acid-active proteases (e.g., pepsin [16] and Protease type XIII from *Aspergillus saitoi* [17]). The resulting peptides are then submitted to LC/MS analysis to allow their molecular weights to be determined. The HDX can be measured by taking the difference between the “average” molecular weight of the deuterated and non-deuterated samples. By comparing the extent HDX of peptides from a protein in bound vs. unbound states, we can identify those regions undergoing conformational change upon ligand binding (the overall workflow is shown in Figure 1A). This binding interface between a ligand and protein is less solvent-exposed upon binding; thus, the HDX at the binding interface is lower (i.e., protection is provided). Conformational change can either expose or bury a protein surface, causing the HDX level to be higher or lower, respectively.

### 2.2. Fast Photochemical Oxidation of Proteins (FPOP)

FPOP employs hydroxyl radicals generated from H_2_O_2_ as the labeling reagent [18,19]. A mixture of H_2_O_2_, a protein solution and a scavenger (e.g., Gln or His [20]) passes through the 248 nm KrF excimer laser window propelled by a flow system, as illustrated in Figure 1B [18]. The hydroxyl radicals formed upon laser irradiation oxidize the protein in a controlled manner upon adjustment of H_2_O_2_ and scavenger concentrations, usually resulting in attaching the mass tags of +16 and +32 (i.e., substitution of one or two H for OH). To ensure that the footprinting occurs faster than protein folding or unfolding, the lifetime of primary reactive hydroxyl radicals is limited to ~1 μs depending on the scavenger’s identity and concentration [18,21], even though the lifetimes of less reactive secondary radicals are in the order of milliseconds in solution [22]. The flow rate of the solution is calculated by considering the laser frequency and laser pulse width to provide an exclusion volume of 15–25% [23] (i.e., a volume that is minimally affected by the laser). The remaining H_2_O_2_ in the solution is then decomposed by methionine and catalase in the collection tube [21].

### 2.3. Specific Amino-Acid Footprinting

Another irreversible labeling strategy is to utilize a reagent that specifically reacts with the side chains of a subset of amino acids in a protein (Figure 1C). Reagents targeting charged residues may be especially useful for studying metal–protein interactions, which rely on electrostatic forces [24]. The reagent usually reacts with the solvent-exposed side chain in the range from minutes to hours, unlike FPOP, where the reactions occur in microseconds [18,21]. After the completion of the labeling, the reagent-responsive and solvent-accessible residues contain a mass tag corresponding to the labeling product. This introduction of the mass tag enables latter localization of the footprinting on the protein by digestion and the tandem MS (MS/MS) analysis of the resulting peptides [25].

For irreversible labeling, the footprinted samples are usually submitted to bottom-up analysis that locates the modified peptides or residues (Figure 1D). The digested peptides with and without modification are often separable by reversed-phase liquid chromatography (LC). The areas under the extracted ion chromatograms (EIC) for the unmodified vs. modified peptides are used to quantify the modification fraction (the ratio of the EIC area of modified peptide and sum of the EIC areas of the unmodified and modified peptides [23]). By comparing the modification fraction in a differential manner (needed to compare different protein states), investigators can identify the binding interfaces or the regions where conformational changes occur. Residue-level information can be provided when modified peptide isomers are separable by LC and the modified residues within a peptide can be assigned unambiguously by MS/MS fragmentation [23].

### 2.4. Native MS

In addition to the footprinting methods discussed above, native MS is a label-free approach for the study of proteins of higher-order structures and labile protein–ligand interactions in the gas phase. Most buffer solutions for structural biology study contain a nonvolatile salt, which interferes with ionization [26]. To compensate, the sample preparation for native MS usually requires a buffer exchange of the protein solution from the MS-incompatible buffer to a non-denaturing and volatile buffer (usually ammonium acetate solution with pH and ionic strength similar to those of the original buffer) [27]. Native MS nearly always uses nano-ESI and gentle ionization to transfer proteins from the aqueous to the gas phase for activation, mass analysis and detection [26,27].

## 3. Qualitative Studies and Stoichiometry Determination by MS-Based Methods

A qualitative study of a metal–protein binding usually lays the foundation for later systematic characterization. To characterize the biochemical properties of a metal–protein system, the steps are to identify the metal that binds to the protein, to estimate its binding affinity, and then to determine the binding stoichiometry. Traditionally, binding stoichiometry is derived from fitting titration data from a binding experiment. Here, we show that native MS allows for direct metal/protein stoichiometry measurement (Section 3.1) Thus, metal binding stoichiometry >1 can be easily examined. MS-based titration methods can afford binding stoichiometry at a global, peptide, or even residue level (Section 3.3). IM coupled with native MS is especially useful when the metal ion/protein complex has different conformations, since IM can separate proteins of different conformers based on their mobility (Section 3.2).

### 3.1. Determining Stoichiometry by Native MS

Native MS can directly answer whether a metal ion binds to a protein and what is the binding stoichiometry. Unlike other biophysical tools with which researchers study binding indirectly through the change in protein stability upon the binding of metal ions, native MS can be utilized to examine binding directly by evaluating the mass changes corresponding to additions of metal ions or other ligands to the protein. The following examples illustrate these capabilities.

#### 3.1.1. Mn^2+^ as Co-Factor for SFTSV Endonuclease

In 2020, Wang et al. used a combination of biophysical tools to delineate the *L*-polymerases cap-snatching mechanism of severe fever with thrombocytopenia syndrome virus (SFTSV) [28]. They demonstrated that the virus hijacks the host mRNA for its own RNA transcription through *L*-polymerase endonuclease activity [29]. Motivated by previous findings [30,31] indicating that divalent metals are necessary for the segmented negative-sense RNA virus’ (sNSV) endonuclease activity, the authors investigated the interaction between the endonuclease domain of SFTSV *L*-polymerase (SFTSV endonuclease) and metal ions [28]. They first employed a thermal shift assay (TSA) to evaluate the endonuclease stability in the presence of Mn^2+^, Mg^2+^ and Ca^2+^ and found that the addition of Mn^2+^, but not Mg^2+^ or Ca^2+^, increased the thermal stability (i.e., the melting temperature increased from 51.8 °C to 54.2 °C) for SFTSV endonuclease, suggesting that only Mn^2+^ binds the SFTSV endonuclease. Consistent with the TSA results, the investigators observed the Mn^2+^–SFTSV endonuclease complex by native MS, and they identified two Mn^2+^-bound SFTSV endonuclease species, one with one bound Mn^2+^ and another with two bound Mn^2+^. They also employed a fluorescence resonance energy transfer (FRET)-based endonuclease assay to show that Mn^2+^ is needed as a co-factor for SFTSV endonuclease activity. These observations are consistent with a previous finding that Mn^2+^ acts as co-factor for other sNSV’s endonucleases [32].

#### 3.1.2. Iron Binding by Sidercalin

Native MS can also be used in conjugation with other MS tools to answer a question encountered in a bigger biological story that otherwise cannot be resolved by using several other biophysical structural tools [33,34]. The investigation of the binding between sidercalin and enterobactin is an example. Sidercalin (Scn) is an anti-microbial protein secreted by host cells during human urinary tract and other infections to withhold iron, a growth-limiting micronutrient for most pathogenic bacteria. The enterobactin (Ent), a secreted microbial, small-molecule chelator (siderophore), promotes pathogenic bacteria growth by chelating ferric ions with high affinity [35,36]. Scn’s capability to sequester iron from E. coli in competition with bacterial Ent depends on urine composition [37]. To study a urine-derived specimen, Shield-Cutler et al. [38] developed a liquid chromatography differential scanning calorimetry (LC-DSF) method to enable the identification of the LC fractions where Scn exhibited an increased melting temperature, indicating a greater ligand occupancy of Scn. Gas chromatography mass spectrometry (GC-MS) identified several potential Scn ligands candidates in active LC-DSF fractions (increased Scn melting temperature). Shield-Cutler et al. sequentially employed DSF and fluorescence quenching (FQ) to assess the binding of 13 commercially available candidates. Among those, pyrogallol, caffeic acid, 3-methylcatechol, 4-methylcatechol, and propyl gallate, all containing a catechol moiety, were double positive. Following a differential GC-MS metabolite analysis, the investigators found pyrogallol and caffeic acid were more abundant in the urine samples with higher Scn activity than in those with lower activity. Then, the investigators used native MS to study the binding of Scn to pyrogallol or caffeic acid. When pyrogallol or caffeic acid were added to the system containing ferric ion and Scn, new species corresponding to Scn/Fe(pyrogallol)_2_ or Scn/Fe(caffeic acid)_2_ appeared. Additionally, when a mixture of pyrogallol, caffeic acid and catechol was added, a species corresponding to Scn/Fe(pyrogallol)(caffeic acid) was seen [38]. This observation, for the first time, supported that Scn used heterogeneous ligands to chelate iron and suggested the role of urinary metabolites as co-factors for Scn to exert its anti-bacterial function.

In 2020, Guo et al. [39] carried out a detailed structural proteomics study to investigate the binding of Scn to Ent with unbound and bound ferric ions. The investigators first used native MS to follow the solution composition after incubating Fe(III)–Ent and Scn. After incubation, they found Scn mostly shifted to higher *m*/*z* with the same charge state distribution; the new peak represented a complex of iron, Ent and Scn in a 1:1:1 ratio. The binding of Ent to Scn in the absence of iron was also confirmed by a peak corresponding to an Ent–Scn (1:1) complex. Even though the binding of hydrolyzed aferric Ent to Scn was reported before [34], Guo’s native MS data provide direct evidence for intact aferric Ent binding to Scn in solution. The native MS data motivated Guo et al. to further characterize the binding site and associated conformational changes by using HDX and specific amino-acid footprinting (Section 4.1 and Section 4.2, respectively).

#### 3.1.3. Ca^2+^ Binding to Centrin

Native MS is especially useful when a protein has different conformations, because this approach can be used to study the metal binding of each conformer in one experiment. This was demonstrated for human centrin-2 (HsCen-2) by Craig et al. [40]. HsCen-2, an EF-hand protein, is essential for cell division as a component of the centriole [41]. The binding of calcium to HsCen-2 is required for assembling the centriole complex [42]. Craig et al. [40] first identified two conformers of monomeric HsCen-2 by observing two charge-state distributions in a native mass spectrum—a lower *m*/*z* range (from 20+ to 10+) for conformer 1 and a higher *m*/*z* (9+ and 8+) for conformer 2. Both conformers formed metal-binding complexes upon the addition of Ca^2+^ or Mg^2+^, but conformer 2 formed a complex with Ca^2+^ and Mg^2+^ ions more readily and completely than conformer 1.

To further investigate which metal ion had higher affinity to HsCen-2, the investigators performed a competition experiment by titrating a preformed Mg^2+^/HsCen-2 complex by increasing the amount of Ca^2+^, or vice versa. For both conformers, Ca^2+^ is more effective at displacing Mg^2+^ than Mg^2+^ at displacing Ca^2+^, indicating that Ca^2+^ bound to HsCen-2 with higher affinity. They also carried out near- and far-UV CD characterization to show that the binding of Ca^2+^ or Mg^2+^ to HsCen-2 had little impact on its structure, consistent with the observation by native MS indicating that the charge state distribution of Ca^2+^- or Mg^2+^-bound HsCen-2 was similar to that of metal-free HsCen-2. Together with the CD melting curve data showing that Ca^2+^ but not Mg^2+^ increased the melting temperature of HsCen-2, the investigators concluded that Ca^2+^ binding stabilized the HsCen-2 structure.

These studies demonstrate that native MS can characterize metal–protein interactions when using low sample amounts (a few µL of low µM) in a direct and rapid way and can screen metal ion binding to a protein. Despite the informative output, debate is still ongoing regarding whether native MS data can accurately represent the protein and its properties in solution. The desalting step during sample preparation can impart structural changes to the protein. In addition, a weak ligand–protein complex may dissociate during the desalting but adding the corresponding weak ligand to the native MS-compatible buffer can help preserve the weak complex. Using a native MS-compatible buffer with high purity is essential for metal-binding proteins given that K^+^ and Na^+^ in solution may bind non-specifically to the protein of interest to give confusing results. To address these issues from an MS perspective, Susa et al. proposed a submicron emitter tip to reduce protein salt adducts [43]. Despite the challenges of tip clogging and increased interaction with the glass surface of the emitter tip [44,45], submicron tips offer promise to make native MS compatible with biological buffers.

Conducting bioassays of the protein in MS-compatible buffers can check that the proteins retain their function. If a native structure is preserved in the buffer, there still are questions regarding whether nano-electrospray and the “gentle” MS source settings retain protein structure while transferring the protein from the solution to the gas phase (i.e., the degree of “nativeness” is still controversial [26]). Hydrophobic interactions are weakened in the gas phase [46] and, thus, complexes with dominantly hydrophobic interfaces may not survive the spray. On the other hand, desolvation can introduce non-specific binding. Metal–protein interactions mainly rely on ionic interactions that are strengthened in the vacuum of a mass spectrometer [47]. The interpretation of native MS data alone requires expertise and care. For non-native MS practitioners, it is recommended to use native MS with other MS footprinting methods or biophysical tools to arrive at a defensible conclusion.

### 3.2. IM-MS for Qualitative Studies of Ca^2+^ Binding to Calmodulin

IM coupling with native MS can offer structural information, including stoichiometry and conformational change at the global level, for conformationally heterogeneous protein samples in the gas phase. Several reviews are available describing IM development, guidelines, and applications in structural studies [48,49,50,51]. IM-MS has been employed to study conformational changes induced by metal ion binding on α-synuclein [52], metallothionein [53], β2-microglobulin [54] and tau protein [55], among others. Here, we show some calmodulin studies as examples.

Faull et al. [56] employed IM-MS to study the binding of Ca^2+^ to calmodulin. They observed two conformations for calmodulin at lower charge states (from +7 to +9) and only one conformation for higher charge states. The collision cross-section (CCS) calculated from the calmodulin NMR structure (1870 Å^2^ [57]) was larger than the conformer with lower CCS, from +7 to +9, suggesting that the IM-MS structure is a native-like structure. Compared with apo calmodulin, the Ca^2+^-bound state showed higher CCS for lower change states (from +7 to +9) but exhibited lower CCS for higher charge states. The binding of a second and third Ca^2+^ to calmodulin further decreased the CCS, indicating a more compact conformation upon binding to Ca^2+^.

Consistent with the finding by Faull et al. [56], Wyttenbach et al. [58] also found that the populations of compact calmodulin increased as increasing amounts of Ca^2+^ bound to calmodulin, reaching the optimal stoichiometry of 4:1 (Ca^2+^: calmodulin). In addition, the investigators showed that peptide ligands, including melittin and CaMKII peptides, formed complexes with Ca^2+^ bound calmodulin in a 4:1:1 (Ca^2+^: calmodulin:peptide) stoichiometry. Significant conformational changes were not observed when peptide ligands bound to 4Ca^2+^-bound calmodulin, suggesting that peptide ligands bind readily to the compact form of calmodulin.

### 3.3. Native Top-Down MS for Study of Metal Ions Binding to Aβ42

The combination of native MS and top-down ion activation (tandem MS fragmentation of an intact protein) without prior proteolysis may enable identification of the ligand binding region for small proteins in the gas phase by monitoring the backbone fragments with the ligand retained [59,60,61]. Lermyte et al. [62] demonstrated this capability for Aβ42 binding with metal ions. Amyloid plaques are a core physiological hallmark of Alzheimer’s (AD) [63] and Parkinson’s diseases. The plaques can be found in AD-infected brains and contain elevated levels of zinc, iron, and copper [64]. To obtain stable spray for a fragmentation experiment, the authors used native MS to optimize the buffer conditions and the metal ion concentration. By fragmenting the 1:1 metal ion to Aβ42 complex using collision-induced dissociation (CID), electron capture dissociation (ECD) and infrared multiphoton dissociation (IRMPD), the authors showed four groups of metal-binding behaviors. The most strongly bound metals, Cu^2+^, Ni^2+^ and Co^2+^, bound to the N-terminus of Aβ42 (~5–14), which contains His. Monovalent and weakly bound K^+^ and Na^+^ interacted with Aβ42 at the C-terminus (~36–40), which lacks residues with polar side chains. Ca^2+^, Mg^2+^ and Mn^2+^ binds at a slightly different regions of the Aβ42 C-terminus (~32–37) with high hydrophobicity and flexibility. Fe^3+^ binds to Aβ42 at residues 8–25, which is composed of Tyr, Glu, His and Asp residues.

### 3.4. Titration Methods for Stoichiometry and Affinity

Two MS-based titration methods were developed by our group. Protein–ligand interactions in solution by mass spectrometry, titration and HDX exchange (PLIMSTEX), by Zhu et al. in 2003 [65], utilizes HDX in a titration format. The extent of HDX is followed as a function of increasing total ligand concentration ([ligand]_total_). Similar to other biophysical titration experiments, such as ITC [66], the shape of the titration curve is sensitive to the total protein concentration ([protein]_total_). As a result, PLIMSTEX can be utilized either to determine binding stoichiometry or to measure binding affinity. For stoichiometry, high protein concentrations (100 times the K_d_) are used. In the resulting plot (“a PLIMSTEX” curve), the ratio between [ligand]_total_ and [protein]_total_ at the sharp break is the stoichiometry of this binding system (blue curve in Figure 2). For binding affinity measurements, a lower protein concentration (~K_d_) is used. The binding affinity can be calculated by fitting the PLIMSTEX curve using a 1:n (protein:ligand) sequential binding model (black curve in Figure 2) [67,68].

LITPOMS, developed by Liu et al. in 2019 [69], shares similar advantages with PLIMSTEX but uses oxidative modifications by •OH as the reporter. Similarly, the binding stoichiometry can be identified when LITPOMS is carried out at high protein concentrations and the binding affinity can be calculated at lower protein concentrations. PLIMSTEX has been used to determine the binding stoichiometry between human calprotectin and Ca^2+^ [70] (details discussed in Section 6).

Compared to traditional biophysical titration methods, MS-based methods use low sample amounts (high picomoles) and provide higher spatial resolution when coupled with bottom-up analysis. Compared to native MS, MS-based titration methods measure protein properties in a relevant aqueous phase. To determine accurate stoichiometry, some HDX experiments need to be conducted in the vicinity of a sharp-break point. Each measurement takes 5–20 min. LITPOMS shares the same drawbacks, given that FPOP-labeled samples often use higher chromatographic resolution in the LC-MS analysis, requiring more time. Both HDX and FPOP data processing requires manual validation. Overall, compared with native MS, MS-based titration approaches require more time; thus, they are not suitable for the fast screening of binding metals.

## 4. Identification of Metal Binding Site and Resulting Conformational Changes

A valuable application of MS-based structural proteomics tools is the determination of the binding site of metal ions and the conformational changes that occur upon binding. The resulting insights are pivotal for understanding the mechanism of binding. Several MS tools can detect the binding site and conformational changes. HDX is the most popular footprinting method owing to its robustness, but it may not be the most effective tool for identifying metal ion binding site because metal ions bind to amino-acid side chains instead of the protein backbone. Irreversible covalent labeling can detect metal ion binding sites with high sensitivity, because this footprinting probes the protein side chains. Targeted amino acid footprinting, designed for charged side chains, is a good starting point for a non-structural proteomics or biochemistry lab, because this footprinting requires no special instrument/lab set-up for the protein labeling process. The stability of the labeled samples also allows for the samples to be shipped to an MS facility for an LC/MS/MS analysis.

### 4.1. HDX-MS

HDX-MS can determine the binding site and conformational changes with high reliability. HDX can be measured at the protein global level or peptide level, depending on sample preparation and processing. Measuring HDX at the global level significantly reduces the data analysis time, offering relatively fast screening and the opportunity to check whether a metal ion is interacting with the protein of interest.

#### 4.1.1. Ca^2+^ Binding to Calmodulin

Nemirovskiy et al. [71] employed HDX-MS to study the binding of metal ions to calmodulin. When 0.04 and 0.49 mM of Ca^2+^ were added to calmodulin solutions, the extent of HDX decreased by 11 and 24 Da compared to Ca^2+^-free calmodulin after 60 min of HDX incubation time, respectively. This suggests that calmodulin adopts a more compact overall conformation when bound by Ca^2+^. In contrast, the extent of HDX was less than 11 Da when 0.49 mM Mg^2+^ was added, suggesting a less compact conformation when bound by Mg^2+^ instead of Ca^2+^. Additionally, lysosome, when tested for Ca^2+^ binding as a control, showed negligible change in HDX upon binding. This work serves as a good example of rapid detection of conformation change upon metal binding by HDX-MS and illustrates a screening tool for qualitative studies.

#### 4.1.2. Ca^2+^ Binding to DREAM

To locate the binding sites in a protein, Zhang et al. [72] integrated global- and peptide-level HDX to study the binding between Ca^2+^ and DREAM (downstream regulatory element antagonist modulator). DREAM regulates transcription by binding to DRE (downstream regulatory element), a transcriptional silencer, in a Ca^2+^ concentration-dependent manner [73,74,75,76,77]. DREAM contains four EF hands [74,76]. The investigators first measured, at the global level, the mass shift of DREAM with and without Ca^2+^ as a function of HDX exchange time. The difference between the bound and unbound states was substantial (~50 Da after 10 s of HDX) and remained constant for longer exchange times. When the global kinetic curve was modeled using five rate-constant bins (100, 10, 1, 0.1 and 0.01 min^−1^) to quantify the Ca^2+^ binding event, the investigators found that 14 of 246 exchangeable backbone amides are fast exchangers (100 min^−1^) for the Ca^2+^-free state. In contrast, for the Ca^2+^-bound state, only 78 exchangeable backbone amides are fast exchangers. The large number of fast exchangeable amides indicates that DREAM has an overall flexible conformation, which becomes more compact and/or undergoes self-assembly upon Ca^2+^ binding. Although the ITC analysis and the NMR structure of N-terminal-truncated DREAM showed that only EF hands 2, 3 and 4 bound Ca^2+^ [78,79], peptide-level HDX revealed that all EF hands of full-length DREAM bound in the presence of Ca^2+^.

To investigate the binding of Ca^2+^ to EF hand 1, the investigators designed two EF hand 1 mutants, for which critical Ca^2+^ binding residues were mutated (E111Q/D112N and E103A/D110A). When Ca^2+^ was present, the extents of HDX of the four EF hands in both mutants were similar to that in the wildtype. Two interpretations can be offered; a mutation in EF hand 1 does not alter the binding property of the other EF-hands, or the protection of EF hand 1 is due to a remote conformational change induced by the binding of Ca^2+^ to other EF hands. Thus, the conflict between ITC and HDX-MS data for the wild-type protein was reconciled by the mutagenesis data.

#### 4.1.3. Ca^2+^ Binding to Human Centrin 2

Sperry et al. investigated the binding of human centrin 2 (HC2) to XPS and Ca^2+^ by using HDX at both the global and peptide levels [80]. As discussed in Section 3.1, HC2 is an EF-hand protein and a structural component of centrioles; it also interacts with Xeroderma pigmentosum group C protein (XPC) and RAD23B to form a DNA damage recognition complex [81]. In this study, the investigators used a Δ25-HC2 construct that lacked self-assembly capability [82] to simplify the experiment. At the global level, HDX shows the protection of XPS –Δ25-HC2 or Δ25-HC2, in the presence of Ca^2+^, is minimal and only at short times (i.e., 15 and 45 s) when compared to the protein in the absence of Ca^2+^. In addition, two populations of Δ25-HC2 with different HDX extents (EX1-like behavior) appeared at 180 s of HDX. Pepsin digestion and an LC/MS analysis showed that the Δ25-HC2 peptide 110–136 (EF hand 3) exhibited EX1-like behavior in the XPC-bound state regardless of the presence of Ca^2+^, suggesting that this region is also involved in XPS binding. The bimodal distribution was the most prominent at 180 s of HDX for the Ca^2+^-free state but shifted, for the bound state, at 600 s. In addition, XPS–Δ25-HC2 peptides 146–162 and 163–172 showed significant protection in the presence and absence of Ca^2+^, whereas Δ25-HC2 peptides manifested little protection. This indicates that the presence of XPS facilitated the adoption of Δ25-HC2 to a conformation that was favorable for Ca^2+^ binding. Peptide 146–162 contains EF hand 4, which binds Ca^2+^ with the highest affinity [83,84]. This study reveals a caveat for global HDX measurements. Even though large HDX differences occurred for the XPS–Δ25-HC2 C-terminus when Ca^2+^ was present, the difference was not substantial and possibly counterbalanced by increases in exposure nearby in the protein.

#### 4.1.4. Zinc Ion Binding to Hepatitis B Virus X

HDX coupled with bottom-up analysis allows the binding site and associated conformational changes to be identified at the peptide level for hepatitis B virus X. Locating binding sites is a common problem in biophysics and was exemplified by Ramakrishnan et al. [85], who employed HDX-MS to study HBx–DDB1 (Hepatitis B virus X protein–DNA damage-binding protein 1) complex in the absence and presence of zinc ions. It is known that a host Smc5/6 (chromosomes 5/6 complex) restricts HBV (Hepatitis B virus) infection by inhibiting HBV genome transcription. To evade host-defense mechanisms, HBV synthesizes HBx to interact with DDB1, leading to the degradation of Smc5/6. Zinc ions interact with HBx, but the details remained elusive [86,87]. The binding of Zn^2+^ to HBx was confirmed [85] when HBx peptides 56–62, 63–71, 111–116 and 132–150 showed protection. To improve the spatial resolution, the investigators also used amino-acid-specific footprinting to pinpoint the residues involved in binding (discussed in Section 4.2).

#### 4.1.5. Iron Binding by Sidercalin

As discussed in Section 3.1.2, Guo et al. [39] utilized native MS to verify the binding of Scn to ferric Ent and aferric Ent. Then, to study peptide-level binding, they used HDX-MS with a statistical analysis to show that regions 34–42, 60–72, 94–109 and 124–139 were protected upon binding Ent (Figure 3A). This result is somewhat different from those obtained from NMR and X-ray crystal studies that showed that R83, K127 and K136 are the ligand binding sites [88,89]. Only peptide 124–139 overlapped with the observed ligand-binding residues K127 and K136. Conversely, the peptide covering R83 showed negligible protection. Peptides 34–42 and 60–72 arise from the β-barrel and their protection suggests a more compact β-barrel in the presence of Ent. For the ferric-Ent–Scn state, peptide 81–94 is from a protected region, covering the R83 residue—the ligand binding interface [88,89]—in addition to the four peptides identified in the differential HDX-MS experiment between Scn and Ent–Scn. The presence of iron enhanced the interaction between Scn and Ent.

#### 4.1.6. Interaction between Cu^2+^ and Aβ42

HDX-MS is effective for characterizing protein aggregation because it offers peptide-level information that cannot be easily obtained by traditional biophysical tools. However, both HDX and aggregation are time-dependent phenomena—both increase with time. To tease apart HDX and aggregation, Zhang et al. coupled pulsed HDX (the HDX labeling time is constant and short (e.g., 1 min) [90,91]) with LC/MS detection to study protein aggregation.

Zhang et al. [92] studied the effect of copper and temperature on the aggregation of Aβ42 by monitoring the protection level of three peptic peptides representing the N-terminus (1–19), middle (20–35) and C-terminus (36–42) regions of Aβ42. The protection levels of the three peptides followed a four-stage sigmoidal increase, consisting of two sigmoids, as a function of the increase in aggregation time at 25 °C. The authors attributed the four stages to the formation of small oligomers from the monomer, the formation of larger oligomers without much structural change, the reorganization of oligomers, and the protofibril formation.

After the introduction of Cu^2+^, the protection, in HDX, did not increase during the aggregation time for all three peptides. The protection level of the N-terminus peptide in the presence of Cu^2+^ was comparable to that of the first plateau stage for the same peptide in the absence of Cu^2+^. In contrast, when Cu^2+^ was added, the protection levels of the middle and C-terminus regions were lower than that without Cu^2+^. Moreover, to further study the effect of Cu^2+^ on the aggregate, the Aβ42 samples with and without Cu^2+^ taken at the longest time were visualized on a native gel. Fewer high molecular weight and more low molecular weight species occurred with Cu^2+^. Based on these data, the investigators proposed that Cu^2+^ stabilizes the soluble Aβ42 lower molecule weight species and prevents the formation of amyloid fibril by interacting with the N-terminal peptide. In addition, Zhang et al. utilized pulsed HDX to investigate Aβ42 aggregation by raising the temperature from 25 °C to 37 °C and incorporating agitation. They found that the lag phase disappeared under accelerated aggregation conditions even in the absence of Cu^2+^, indicating that the conversion between low and high molecular weight species become faster at higher temperatures. When Cu^2+^ was added to the current system, the lag phase reappeared, consistent with the observation, at 25 °C, that Cu^2+^ slowed down the aggregation.

This pulsed HDX/LC/MS platform offers an opportunity to study protein structural change as a function of time with peptide-level spatial resolution. It can be adapted to the study of protein folding/unfolding, oligomerization and aggregation from different perspectives. Intermediate folding can be revealed, kinetic information can be extracted, and the effect of a perturbation to the protein system (e.g., introducing a small molecule, metal ion, peptide, or ligand, or increasing the temperature) can be investigated. Similarly, the platform can provide thermodynamic and protein stability information by following the pulsed HDX reaction as a function of the temperature, concentration of a chaotropic reagent, or pH.

In both manual and robot-assisted HDX experiments, the shortest HDX labeling time is several seconds, limiting pulsed HDX and MS to fast folding/unfolding or aggregation. This limit can be addressed by footprinting with FPOP where the highly reactive radicals react within microseconds. Another approach for HDX at shorter times incorporates a microfluidics such as that developed by Rob et al. [93]. In addition to time resolution, spatial resolution is also a challenge for the usual HDX-MS bottom-up experiment. Traditional HDX relies on acid-resistant proteases to generate peptides that are identified and characterized by MS. The more overlapping peptides are generated, the higher the spatial resolution. Therefore, spatial resolution is associated with protease digestion efficiency and limited by the small number of acid-resistant proteases. One way to improve the spatial resolution, even to the residue level, without relying on proteolysis is to fragment the peptide backbone to locate the residue undergoing exchange. Although CID is a classic fragmentation method and available on most commercial instruments, it fragments an ion by increasing its internal energy through collisions with neutral atoms or small molecules on a time scale from µs to ms [94,95]. Unfortunately, CID fragmentation leads to the scrambling of H and D, to redistribute them within the peptide before fragmentation, preventing the location of the original D. To address this issue, ETD (electron transfer dissociation), ECD and UVPD (ultraviolet photon dissociation) fragmentation can be incorporated in the HDX MS/MS analysis [95,96,97,98]. These fragmentation approaches minimize scrambling, achievable because ETD and ECD occur rapidly (psec) and UVPD deposits high energy into a peptide bond [95], favoring simple cleavage over rearrangement (scrambling).

### 4.2. Targeted AminoAcid Footprinting

#### 4.2.1. Iron Binding by Sidercalin

The structural and HDX-MS (Section 4.1.5) data suggest that the binding of Scn to Ent relies on Lys and Arg. To pinpoint the binding residues, Guo et al. [39] used ethyl acetimidate hydrochloride (ETAT) and methyl glyoxal (MG) reagents to footprint Lys and Arg, respectively. The primary amine group in Lys reacted with the acetimidate group in ETAT via an S_N_2 reaction, leaving a mass tag of +41.0265 Da for MS detection. The dicarbonyl group in MG reacted with the guanidino group in Arg, producing a cyclic adduct with a mass shift of +72.0211 Da or +54.0105 Da (water-loss product) [99,100,101]. After optimizing the labeling to occur at ~ 5 min, the investigators found that R83, K127 and K136 show decreased solvent accessibility (*p* < 0.01) upon Ent and ferric-Ent binding (Figure 3B). R74, R132 and R142 also show >50% difference between the bound and unbound states but with less confidence (*p* > 0.01). An SASA analysis of these residues in the Ent-bound and -unbound states revealed that R74 and R142 became less solvent-accessible in the bound state, whereas R132 displayed no change, indicating R74 and R142 underwent conformational change upon Ent binding. Utilizing HDX-MS, specific labeling and native MS, Guo et al. [39] confirmed the binding of Scn to ferric and aferric Ent. They also showed that the binding of ferric and aferric Ent shared the same binding interface and that the presence of iron enhanced the binding between Ent and Scn. The traditional model of bacterial defense through Scn is that Scn sequesters iron from pathogenic bacteria. Taken together, the results support a model in which Scn combats bacterial infection by hijacking the bacterial iron-chelating ligand Ent.

#### 4.2.2. Zinc Ion Binding to Hepatitis B Virus X

In the same study of HBx reported by Ramakrishnan et al. [85], *N*-ethylmaleimide (NEM) was utilized to footprint the HBx protein in a labeling time-dependent manner. NEM specifically labels cysteine by a Michael addition reaction [102,103,104]. Consistent with the HDX-MS results (discussed in Section 4.1.1), C61, C69, C115 and C137 of HBx showed reduced NEM footprinting in the presence of Zn^2+^. The investigators also tested whether these cysteine residues were essential for HBx function by expressing the corresponding cysteine mutants (C61A, C69A, C115A and C137A) in PHH (primary human hepatocytes) and examining the Smc6 levels. They found that the mutation of C61, C69, C137, but not C115, impaired the HBx function, indicating that only the three cysteine residues were Zn^2+^ binding sites. Given that ligand binding and conformational change can both decrease HDX and specific labeling and that C115 is not a Zn^2+^ binding site, the protection at C115 is likely due to remote conformational change induced by Zn^2+^ binding.

#### 4.2.3. Benzhydrazide Targeting Glu and Asp: Metal Ion Binding to Calmodulin

Proteins bind metals directly through Cys, His, Glu and Asp, depending on the metal’s coordination properties. Metal ions can be classified into (1) soft Zn^2+^, Cu^2+^ and Fe^2+^, that bind to His and Cys residues, containing nitrogen and sulfur as binding sites; and (2) hard, Ca^2+^ and Mg^2+^, that bind favorably to Glu and Asp, containing side-chain oxygen [24]. Specific labeling may be an appropriate tool to study metal/protein interactions, especially when a reagent directly targets the residues that metal ions bind. An example is a reagent targeting the carboxyl group. An effective method to footprint a carboxyl group is to activate it using EDC (1-ethyl-3-(3-(dimethylamino)propyl)carbodiimide) and then attack it with a nucleophile to form a stable product (Figure 1).Glycine ethyl ester (GEE) is one of the most popular choices for the nucleophile, attaching a mass tag of +85 Da to Asp and Glu [105]. It has been employed to study apolipoprotein E, vascular endothelial growth factor, FMO antenna protein, etc. [106,107,108,109,110,111]. Despite the popularity, it requires a high GEE-to-protein ratio, and the labeled product undergoes some hydrolysis during workup and the LC analysis.

To overcome these drawbacks, Guo et al. [105] implemented BDH (benzhydrazide) as the nucleophile, instead of GEE, and coupled isotope encoding for accurate identification and quantification. The labeled Glu and Asp acquired a mass tag of +118.0531 Da (Figure 1), whereas the isotope-encoded version (BHD-*d5* with five Ds on the phenyl group) gives a +123.0845 Da mass shift of the product. When used for the footprinting of calmodulin binding to Ca^2+^ and Mg^2+^, the investigators defined a modification extent difference greater than 50% and a *p*-value less than 0.01 as significant. Significant protection, from the comparison of the Ca^2+^-bound and Ca^2+^-free states, occurred for EF hand 1 (14–21, 22–30 and 31–37), EF hand 2 (38–74), linker region (76–86), EF hand 3 (95–106) and EF hand 4 (127–148), consistent with binding surmised by NMR for unbound calmodulin [57] and the crystal structure of Ca^2+^-bound calmodulin [112].

Whether Ca^2+^ and Mg^2+^ share a binding site for calmodulin has been controversial from the 2010s. A comparison of modification extents of the Mg^2+^-bound and Mg^2+^-free states showed that the binding sites of Ca^2+^ and Mg^2+^ largely overlap, but that the binding details are different. The peptide covering E31 in EF hand 1 in the Mg^2+^-bound state was modified by BHD (0.008%), whereas that in the Ca^2+^-bound state was not detectable. This agrees with evidence indicating that, in metal-bound calmodulin structures, E31 coordinates with Ca^2+^ but is too far away to chelate Mg^2+^ [112,113]. In contrast, peptide 38–74 covering EF hand 2 showed similar labeling for the Mg^2+^- and Ca^2+^-bound states, even though the crystal structure indicates that Ca^2+^ binds to EF hand 2 directly and that Mg^2+^ interacts through water molecules. Although the crystal was not informative on Mg^2+^ binding at the C-terminus, BHD footprinting showed less modification for the linker region, EF hand 3 and EF hand 4 for Mg^2+^ than for Ca^2+^, confirming that Mg^2+^ interacted with calmodulin to a lesser extent than Ca^2+^.

#### 4.2.4. DEPC Footprinting of Cu^2+^ and Zn^2+^ Binding Sites

In addition to the reagents discussed above, ethylene glycol bis(sulfosuccinimidylsuccinate) (DEPC), although able to label several nucleophilic residues via an S_N_2 reaction [114], is the most widely used His-labeling reagent, attaching a mass tag of +72.021 Da to His [12]. Narindrasorasak et al. [115] used DEPC to locate the Cu^2+^-binding His101 and His134 of the copper metabolism gene MURR1 (mouse U2af1-rs1 region1) domain. DEPC was also used to pinpoint the Cu^2+^ [116] and Zn^2+^ [117] binding residues on a prion protein and Zn^2+^-binding sites on α-crystallin [118].

#### 4.2.5. Cross-Linking: Ca^2+^ Mediated Calmodulin–bMunc13-2 Interaction

Chemical cross-linking is a subgroup of targeted aminoacid footprinting. Chemical cross-linkers are bifunctional derivatives that can covalently label two targeted amino acids. Thus, cross-linking/MS (XL/MS) can map the amino acids within a certain distance depending on the cross-linker spacer’s length. In the context of metal ion/protein interactions, Piotrowski et al. [119] used XL/MS to characterize the Ca^2+^-mediated calmodulin–bMunc13-2 interaction. bMunc13-2 is one of the Munc13 isoforms that plays an important role in synaptic plasticity via the Ca^2+^/calmodulin-dependent pathway in the brain [120]. All Munc 13 isoforms share a conserved C-terminus, containing C1, central C2, MUN and C-terminal C2 domain and containing a canonical calmodulin binding site that is positioned before the C1 domain [121].

To characterize calmodulin–bMunc13-2, the investigators first utilized photo-Met chemistry to screen for the optimal Ca^2+^ concentration for complex formation (bMunc13-2 367-780, spanning the putative (572–594 [121]) and the canonical calmodulin binding domain (719–742 [121]), was used). The Met in calmodulin was replaced with photo-Met containing diazirine that can form linear diazo (~75%) or a carbene (~25%) via UV-A activation, and label proximal carboxylic acids and insert into C-H bonds, respectively [122]. The resulting cross-linked calmodulin–bMunc13-2 was visualized on SDS-page. The authors found that 750 nM of Ca^2+^ was enough for the formation of a calmodulin–bMunc13-2 complex at a 1:1 stoichiometry. Therefore, 750 nM and 1 mM Ca^2+^ (a commonly used excess concentration) were used for XL experiments. In addition to photo-Met, the investigators also employed amine-reactive bis(succinimidyl)suberate (BS3), disuccinimidyldibutyl urea (DSBU) and amine-/thiol-reactive *N*-γ-maleimidobutyryloxysulfosuccinimide ester (s-GMBS) to show that 16 unique intermolecular cross-link pairs between calmodulin and bMunc13-2 formed at 1 mM Ca^2+^, with 10 out of 16 pairs being also found at 750 nM Ca^2+^. The identified XL amino acids are not only located within the canonical calmodulin binding domain but also involve the putative calmodulin binding domain and flanking amino acids before the canonical binding domain. Using a combination of cross-linkers targeting different amino acids allows new protein/protein interaction interfaces to be identified. When choosing cross-linkers, investigators need to consider several factors, including amino acids of interest and targeting distance. We do not review details of XL/MS here, since Piersimoni et al. [123] recently published a comprehensive review regarding XL/MS for studying protein higher-order structures.

To accurately translate footprinting data to structural information, the labeling process must not disrupt the protein structure. In other words, the protein of interest must stay in a native-like state during the labeling process. One concern with residue-specific labeling is that its long reaction time (from minutes to hours) possibly induces conformational change to the protein and leads to misleading footprinting results. This effect could be reagent- or protein-dependent. Therefore, the evaluation of protein “nativeness” after labeling is recommended, especially when performing specific labeling. The “nativeness” can be examined by an enzymatic activity check if the protein is an enzyme, and by higher order structure checks by CD, DLS, native MS, etc. Zhang et al. [124] demonstrated the evaluation of “nativeness” of three model proteins—calmodulin, β-lactoglobulin and troponin C—after GEE labeling by using CD and global HDX measurements. An ideal method for checking protein “nativeness” should be relatively fast and sensitive to secondary structure change. The development of a “perturbation suite” of tools to evaluate the “nativeness” after labeling has been an active and ongoing research area in our group, and the problem has not been resolved.

### 4.3. Hydroxyl Radical Footprinting of Calmodulin

Zhang et al. [125] utilized FPOP to study the effect of different ligands on the binding of calmodulin and Ca^2+^. Three peptide ligands—melittin (Mel), mastoparan (Mas) and M13—were investigated. Although the NMR structure of the M13/Ca^2+^–Calmodulin complex was available, the structural information for the binding of Mel and Mas to calmodulin was missing at the time of the study [126]. The investigators asked if the binding of Mel and Mas to calmodulin is similar to that of M13. The peptide-level differential FPOP data from comparing the Ca^2+^-bound and Ca^2+^-free states revealed three peptide behaviors in the absence and in the presence of different peptides. The first group of peptides, represented by 31–37 and 76–86, were not involved in peptide binding, as evidenced by a constant modification extent. The second group comprised 14–21, 22–30, 107–126 and 127–148, showing opposite differential modification extents in the presence and absence of peptides. The last group, consisting of 1–13 and 95–106, did not follow a trend for the modification extent across different conditions. The investigators also employed spectral contrast angle [127] to quantitatively analyze the peptide modification trend reflected by spectral mass and intensity differences and concluded that the three peptides shared similar binding patterns. In addition, residue-level FPOP modification extents were obtained for 14 amino acids and correlated with the reported calmodulin structures [67,112,126]. Residues L18, F19, M109, M124, M144 and M145 showed protection when a peptide was added. These residues are consistent with the interface between M13 and calmodulin in the NMR structure [126]. Of note, Y99, a sensitive reporter for small structural change owing to its proximity to a flexible loop and orientation towards solution, exhibited exposure in the presence of M13 but protection in the presence of Mel or Mas. Overall, Mel and Mas bind similarly to M13 in interactions with Ca^2+^-bound calmodulin, demonstrating that FPOP can provide ligand binding information by comparing modification levels of unknown ligands with a reference ligand with known binding.

## 5. Determining Affinity and Binding Order by MS Titration Methods

Binding affinity is a key component for understanding a binding event. The determination of affinity is especially challenging when the binding stoichiometry is greater than 1. This is often seen in the binding of signaling proteins to metal cations [128]. One approach to measure the regional binding affinity is to design truncated or mutated constructs and determine the binding affinity of an individual domain. However, this method is limited, because it ignores cooperativity, a common feature in the binding of signaling proteins. In contrast, MS-based titration methods allow the regional binding affinity as well as metal ion binding order to be determined in one experiment without altering the native protein. In this section, we describe this capability using calmodulin as an example (an additional example is presented in Section 6).

### 5.1. Protein–Ligand Interactions in Solution by Mass Spectrometry, Titration and HDX Exchange (PLIMSTEX)

Zhu et al. [129] titrated 15 μM calmodulin with 0–0.4 mM Ca^2+^ in a buffer containing 50 mM HEPES (pH 7.4), fit the global PLIMSTEX curve using a sequential-binding model [57,102], and calculated the stepwise macroscopic binding constants, *K*_3_ and *K*_4_. The resulting *K*_3_ (7.12 × 10^4^ M^−^^1^), *K*_4_ (1.10 × 10^5^ M^−^^1^), *K*_1_ (2.51 × 10^5^ M^−^^1^ from Ref. [130]) and *K*_2_ (5.01 × 10^6^ M^−^^1^ from Ref. [130]) were used to model the fraction of each species in the system, revealing that, due to positive cooperativity, 2Ca^2+^-calmodulin and 4Ca^2+^-calmodulin were the most abundant species as the titration proceeded. Increasing the solution ionic strength and salt composition weakened the binding.

The cation identity, rather than the anion identity, in the buffer solution, has an impact on the binding of Ca^2+^ to calmodulin. Calmodulin binds favorably to Ca^2+^ > Mg^2+^ ≫ K^+^. Sperry et al. [131] further improved PLIMSTEX by being the first to incorporate a bottom-up analysis for Ca^2+^ and calmodulin binding as an example. The accuracy of PLIMSTEX curve fitting, evaluated by comparing the root-mean-square (RMS) of the residuals in fitting the global- and peptide-level titrations, was 0.16 and 0.40, respectively, indicating that the former model gave a better fit.

### 5.2. Protein–Ligand Interaction by Ligand Titration, Fast Photochemical Oxidation of Proteins and Mass Spectrometry (LITPOMS)

Liu et al. [132] investigated the binding between Ca^2+^ and calmodulin by LITPOMS. In their study, 1 μM calmodulin was titrated with increasing concentrations of Ca^2+^, ranging from 0 to 60 equiv. The tryptic peptide-level modification fractions were determined, and their behaviors grouped into four categories (Figure 4A). Peptides 76–90 and 107–126, representing Class I behavior, exhibited increased modification, whereas that of Class II, reflected by peptides 1–13 and 31–37, remained relatively constant throughout the titration. Increased protection occurred for peptide 14–30, comprising Class III behavior. Class IV behavior, consisting of peptides 38–74, 91–106 and 127–148, was composite, owing to a combination of binding and conformational changes.

The investigators also extracted the binding order of Ca^2+^ with the four EF hands by comparing the onsets for increased protection. In other words, preferential Ca^2+^ binding causes its EF hand to show decreased modification at lower metal-to-protein ratios. The binding orders are EF hand 4 (127–148) > EF hand 3 (91–106) > EF hand 2 (37–74) > EF hand 1 (14–30) (Figure 4B). The investigators further fitted the peptide-level modification fraction curves for each EF hand and calculated the binding affinities for EF hands from 1 to 4 to be 2.9 × 10^6^ M^−^^1^, 4.1 × 10^4^ M^−^^1^, 6.2 × 10^6^ M^−^^1^ and 1.4 × 10^6^ M^−^^1^, respectively. The success of fitting was demonstrated by the reasonable agreement with the literature binding affinities [130].

To further dissect the complex behaviors of these modification curves, Liu et al. used a different enzyme (chymotrypsin) and MS/MS to improve spatial resolution [133]. Consistent with the peptide level, residues M109 and M124, representing Class I behavior, showed increased exposure with the increase in Ca^2+^. For Class II behavior, only M36 was observed in peptide 31–37; thus, the peptide-level modification was taken as the indicator. F16, F19 and K21 in peptide 13–40 underwent increased protection—Class III behavior—as calmodulin became saturated with Ca^2+^. Class IV behavior is the most complicated. Each Class IV peptide contains multiple residues experiencing either protection or exposure, depending on their location. The use of chymotrypsin in addition to trypsin to generate different peptides can tease apart these behaviors.

The tryptic peptide 127–148, spanning the calcium loop and F-helix in EF hand 4 [57,112], showed decreased modification, followed by increased modification across titration. The rationalization is that Ca^2+^ binding to the EF-hand loop results in protection, whereas binding of Ca^2+^ to the F-helix leads to the exposure of the side chain. Generated by chymotrypsin digestion, peptide 125–138 (calcium loop) showed increased protection, whereas 139–145 (F-helix) lost protection upon addition of Ca^2+^. Similarly, F92 and chymotryptic peptide 90–99 in the calcium loop of EF hand 3, related to tryptic peptide 91–106, underwent increased protection. Despite the lack of a chymotryptic peptide covering the 38–74 region (EF hand 2 and linker between EF hand 1 and EF hand 2), the MS/MS of a tryptic peptide revealed that M71 and M72/R74 underwent increased modification upon binding. M51, in this region, in contrast, still showed increased and then decreased modification.

Overall, LITPOMS reports the binding site, remote conformational change, binding affinity, and binding order of Ca^2+^ in one experiment as one explores the protein from one end to the other. Spatial resolution is the key to improve either the fitting outcome or resolving composite behaviors in PLIMSTEX and LITPOMS. As discussed at the end of Section 4.2, improving the spatial resolution of PLIMSTEX requires a combination of acid-resistant proteases and scrambling-resistant MS2 fragmentation. The LITPOMS bottom-up analysis can employ a wider choice of proteases, owing to the irreversible labels introduced in the peptide.

## 6. Illustration of Integrated Methods: Ca^2+^ Binding to Calprotectin

Adhikari et al. [70] used an integrated MS platform—HDX-MS, PLIMSTEX and native MS—to delineate the complex binding between Ca^2+^ and human calprotectin (hCP). hCP serves as an anti-microbial protein by sequestering essential transition metals in response to microbial invasion [134]. Binding of Ca^2+^ increases hCP’s binding to transition-metal ions to deny them to the microbe, thereby enhancing anti-microbial activity [135]. Small hCP contains S100A8 and S100A9 subunits, each of which contains two Ca^2+^ binding motifs—a C-terminal canonical and an N-terminal non-canonical EF hand [136]. As a result of Ca^2+^ binding, the S100A8/S100A9 dimer undergoes tetramerization [137]. However, the X-ray crystal structures of metal-bound tetrameric CP-Ser (a disulfide bond-free mutant) showed that CP-Ser bound fewer than three Ca^2+^ [138,139].

HDX-MS was employed to examine the CP-Ser structure in the presence and absence of Ca^2+^ [70]. Consistent with a prior crystallography study [138,139], the protection of the C-terminus EF-hand regions of both subunits (e.g., S100A8 55–68 and S100A9 65–77) was higher than that of the N-terminus (e.g., S100A8 16–26, 27–34 and S100A9 22–36), suggesting that Ca^2+^ bound more tightly to the C-terminus. The hydrophobic residues—S100A8 I60, L72, L74 and S100A9 W88—formed a tetramerization interface in the crystal structure [138]. In contrast, protection only occurred for S100A8 58–68 and 73–93 and not for S100A9 87–114. Moreover, the C-terminal tail of S100A9 (96–114), which binds transition metals in the crystal structure [138], showed no HDX change in the presence of Ca^2+^, indicating that the enhanced binding capability of CP-Ser with transition metals in the presence of Ca^2+^ is not due to induced conformational change.

The investigators used PLIMSTEX to obtain another view and monitored the Ca^2+^ binding regions and the CP-Ser tetramerization interface pinpointed by HDX-MS (Figure 5A). In the PLIMSTEX experiment, 4 μM CP-Ser was titrated with increasing concentrations of Ca^2+^ up to 500 μM, revealing that the EF-hand regions of both subunits (S100A8 27–34, 55–68 and S100A9 23–37, 66–78) were protected by the introduction of Ca^2+^ and that S100A8 peptides covering or adjacent to the tetramerization interface (55–68, 75–93) showed protection, whereas that of the S100A9 subunit (87–113) did not. PLIMSTEX under sharp-break conditions (Figure 5B) showed that increasing concentrations of Ca^2+^ up to 480 μM (i.e., 12 equiv.) for 40 μM CP-Ser gave 4:1 break (Ca^2+^/CP-Ser) for peptides 27–39 and 35–45, covering the S100A8 N-terminal EF hand; peptide 9–13, at the S100A9 N-terminus; and peptide 61–65, adjacent to the C-terminal EF hand of S100A9. However, peptides 54–62, covering the C-terminal EF hand of S100A8, and 14–21, adjacent to the N-terminal EF hand of S100A9, broke at 3:1 or 2:1 stoichiometry. The results demonstrate that the conformational changes in the C-terminal EF hand of S100A8 and the peptide adjacent to the N-terminal EF hand of S100A9 occurred earlier. The overall sharp-break PLIMSTEX results indicate that heterodimeric CP-Ser bound 4 equiv. of Ca^2+^ and that the heterotetramer bound 8 equiv. of Ca^2+^.

Native MS was also employed to measure the binding stoichiometry (Figure 5C). Measurements were taken for 15 μM CP-Ser when Ca^2+^ was added at several points, ranging from 0 to 500 μM. The abundance of tetramers increased whereas that of dimers decreased with the increase in Ca^2+^. The transition from dimers to tetramers seemed to occur at 200–300 μM (i.e., 13–20 equiv.) of Ca^2+^. When the concentration of Ca^2+^ reached 200 μM, the species corresponding to tetramer-8 Ca^2+^ appeared. These data indicate that the dimeric CP-Ser with 4 Ca^2+^ bound was the preferred species to form tetramers. A 1:1 tetramer:dimer appeared at 225 μM (i.e., 15 equiv.) of Ca^2+^, as shown in the bottom panel of Figure 5C.

The data, taken together in this integrated MS approach, show that Ca^2+^ binding induces the dimeric CP-Ser self-association into a tetramer, which binds eight Ca^2+^ and forms a transition-metal binding site. To quantify the binding, the PLIMSTEX data of four EF-hand-containing peptides were fit using a 4:1 (Ca^2+^:dimeric CP-Ser) sequential binding model with incorporation of the tetramerization and cooperativity to obtain four different models that cannot be distinguished. Sharp-break PLIMSTEX and native MS titration data constrained the models and the calculation of the four binding constants for Ca^2+^ (*K*_1_ = *K*_2_ = 9.4 × 10^−7^ M^−1^, *K*_3_ = 4.9 × 10^−5^ M^−1^, *K*_4_ = 4.3 × 10^−1^ M^−1^) and a tetramerization constant (*K*_tetramer_ = 1.1 × 10^−12^ M^−1^). Based on the HDX-MS data and the X-ray crystal structure, *K*_1_ and *K*_2_ were assigned to the C-terminal EF hands that bound Ca^2+^ more tightly.

The MS data support the current working model in the field, that the conformation of hCP is Ca^2+^ dependent. hCP exists as a Ca^2+^-free dimer in the cytoplasm where [Ca^2+^] ~ 100 nM. It transitions to a tetramer and acquires higher binding capability for transition metals when it is secreted into extracellular space, where [Ca^2+^] ~ 2 mM [134,140,141]. The MS data also provide additional details for this model, including the amount of Ca^2+^ needed for the transition from dimer to tetramer, the binding stoichiometry between Ca^2+^ and hCP, the binding affinity between Ca^2+^ and hCP, and the affinity between S100 subunits. Despite the success of this work, the cooperativity, often seen in signaling proteins, could not be identified [140]. Moreover, some ambiguity exists owing to limited spatial resolution. For example, residue W88 in the S100A9 subunit was part of the tetramerization interface in the crystal structure, but the S100A9 peptide 87–114 showed no change when Ca^2+^ bound. This is likely due to the dilution effect often seen in long peptides encountered in bottom-up HDX-MS (the differential deuterium uptake level is diminished by the large number of residues with no change in HDX). As discussed in Section 3.2, another enzyme or MS2 fragmentation may be utilized to improve the information content of the data.

## 7. Conclusions and Outlook

We reviewed the applications of MS-based footprinting approaches to study the interactions between metal ions and proteins. The studies reviewed include qualitative ones, others focused on stoichiometry, binding site, conformational change, binding affinity, and binding order. We chose examples principally from our lab to illustrate how MS can bring understanding to the binding between proteins and metal ions.

Efforts continue to develop new footprinting tools and improve the current tools for probing protein ligand interactions. Labeling reagents, sample processing, data acquisition, and data analysis have all been improved. New reagents are being proposed to target different amino acids that cover a wider range of residues involved in binding. Increased spatial resolution, from the global protein level to the peptide level and, ultimately, residue level, allows binding sites and conformational changes to be precisely identified with spatial resolution.

Owing to irreversible labeling, high spatial resolution is relatively easier to obtain from off-line digestion and a commonly available fragmentation technique such as CID and HCD than what can be achieved with HDX-MS. The rapid evolution of separation techniques and hybrid mass spectrometers enables the efficient quantification and location of labels in footprinting. The evolution of reversed-phase chromatography and the incorporation of ion exchange chromatography (IEX) and capillary electrophoresis (CE), among others, facilitates the separation of peptides and peptide isomers, thus improving the identification and dynamic range [142,143]. These separation methods have also been coupled to native MS to eliminate the tedium of off-line buffer exchange, which may lead to structure perturbation [144,145]. Data analysis software improvements give more reliable peptide identification and shorten the analysis time. Data processing software is clearly more mature for HDX than for irreversible labeling. Several HDX-MS software packages permit peptide-level kinetic curves and global protein heat maps to be output [146,147,148]. Statistical analyses for processing HDX-MS data to establish differences, equivalence, and detection limits are now being described [149,150,151], which are not yet available for irreversible labeling.

In conclusion, the best approach to metal/protein binding is to utilize a combination of different MS-based and other biophysical tools to provide complementary information, as shown for calprotectin in Section 6. For example, footprinting locates binding interfaces and regions undergoing conformational change upon binding, and MS titrations give stoichiometry and affinity. However, footprinting alone cannot distinguish changes due to binding vs. remote conformation changes. X-ray crystallography and CryoEM do not readily capture conformational changes or dynamics, but they do provide insights into binding interfaces. By choosing a productive combination of methods, investigators will be able to answer more readily the complex issues that continue to arise in understanding protein/metal binding.

## Data Availability

Not applicable.

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
