# Peer review of "Mass Spectrometry-Based Structural Proteomics for Metal Ion/Protein Binding Studies"

_biomolecules, 2022, doi:10.3390/biom12010135_

Round 1

Reviewer 1 Report

The review by Lin and Gross is overall well written but would benefit from some reordering and inclusion of work by other groups. The order of topics in the manuscript currently does not always flow well. The text jumps a bit between methods and proteins.

The authors may want to capitalize on the introduction of methods that they provide and then describe the combinatory benefit of the methods on a few examples similar to chapter 6, e.g. 4.1.5 and 4.2.1 should be combined. However, chapter 6 could be a bit more concise. Additionally, some methods are seemingly introduced several times, i.e. repetitive phrasing appears. Through the suggested rearrangement this review could potentially also be sharpened and a more profound conclusion section provided.

The review would also benefit from presenting ion mobility alongside native MS. It is currently only mentioned in a few lines towards the end of the manuscript.

Although the authors say and intentionally discuss mainly their own work, the review would greatly benefit from including work by other groups. The focus on own work is uncommon and does not allow for putting results into more general perspective. Many other groups have looked into protein-metal ion interactions by structural MS, e.g. calmodulin.

The review would benefit on a short discussion of protein-ligand binding in native MS in general, which needs to be carried out carefully to avoid artefacts. Luckily, this is less of an issue with high affinity metal binding then with other weak ligands. Nevertheless, this would be useful for the non-expert. The statement towards the end of page 6 is a bit too brief to cater for this.

More specific comments:

- l.48-50: I would be good to also discuss contributions by others. Many labs employ covalent labelling, also data on metal binding proteins.

- l.69ff: HDX should be used as abbreviation for the exchange reaction not the entire process including MS, which is used in a confusing fashion in the following text.

- l.94: How about the work from Konermann et al. showing that radical reactions carry on much longer despite scavenger addition?

- authors should check whether sufficient references are provided in section 2, the referencing appears scarce for certain paragraphs.

- 3.1.1. title is a bit confusing is cap-snatching by viral polymerases meant? Only some of these are called L protein. The need for Mn2+ was reported earlier for L proteins, e.g. in Holm et al 2018 JBC. The way it is written now, it suggests that 2020 Wang et al are the first to show this.

- l.174ff: mutation to what? Alanine?

- 3.2 Abeta metal binding has also been analysed by native MS, e.g. by Young et al, it would be useful to present this alongside.

- l.354/355: I agree that HDX and FPOP are superior in localizing binding but doubt that these are superior in determining stoichiometry as they will be subject to similar issues as ITC for stoichiometry determination. Please rephrase slightly.

- l.364: strictly speaking HDX is not label-free, it just employs the smallest possible label.

- 4.1.1.: if not mistaken Calmodulin metal binding has also been investigated by native MS, oxidation profile and various other techniques. It would hence be a good example to show the contribution of the various techniques and also showcase work from other groups.

- l.399: only half of the mutation sites are exchanged for alanine.

- l.514: introduce GEE

- scheme 1: the link to the protein is different on left and right side lacking a carbon atom prior to the carbonyl group on the right side.

- l.556-558: also many cross-linking groups have looked into that. Citing some manuscripts here would provide a more balanced review.

Author Response

Dear Editors and Reviewers of Biomolecules:

We sincerely thank you for your insightful comments on our manuscript entitled “Mass Spectrometry-based Structural Proteomics for Metal Ion/Protein Binding Studies”. We made revisions based on the comments from all reviewers, and we feel that this has significantly improved the manuscript.

Besides modifications done in response to the reviewers’ remarks, we did a complete editing of the manuscript to repair any sections needing clarification and grammar improvements.

We detail our responses to the reviewers’ queries and suggestions below. Please do reach out to us if we can clarify or provide any further information which may be helpful. We sincerely look forward to hearing from you regarding our revised manuscript.

Reviewer 1

Comments and Suggestions for Authors

The review by Lin and Gross is overall well written but would benefit from some reordering and inclusion of work by other groups. The order of topics in the manuscript currently does not always flow well. The text jumps a bit between methods and proteins.

The authors may want to capitalize on the introduction of methods that they provide and then describe the combinatory benefit of the methods on a few examples similar to chapter 6 (calprotectin), e.g. 4.1.5 (sidercalin hdx) and 4.2.1 (sidercalin covalent labeling) should be combined. However, chapter 6 could be a bit more concise. Additionally, some methods are seemingly introduced several times, i.e. repetitive phrasing appears. Through the suggested rearrangement this review could potentially also be sharpened and a more profound conclusion section provided.

The review would also benefit from presenting ion mobility alongside native MS. It is currently only mentioned in a few lines towards the end of the manuscript.

Although the authors say and intentionally discuss mainly their own work, the review would greatly benefit from including work by other groups. The focus on own work is uncommon and does not allow for putting results into more general perspective. Many other groups have looked into protein-metal ion interactions by structural MS, e.g. calmodulin.

The review would benefit on a short discussion of protein-ligand binding in native MS in general, which needs to be carried out carefully to avoid artefacts. Luckily, this is less of an issue with high affinity metal binding than with other weak ligands. Nevertheless, this would be useful for the non-expert. The statement towards the end of page 6 is a bit too brief to cater for this.

Thank you for your comments. We wrote this review as a tutorial of MS tools and not as an “encyclopedic” review, and we made this quite clear in the text. The justification is the audience for the special issue consists of persons not familiar with mass spectrometry. This audience, unlike MS readers, are not interested in all the details on the evolution of the subjects. Thus, we organized the contents by the metal ion/protein binding questions and tools that can be used to address these questions. Nevertheless, we did add new, key references to provide more balance.

Specifically, we added ion mobility section (section 3.2) and top down-down analysis (section 3.3) with the native MS discussion, DEPC footprinting application (section 4.2.4), cross-linking application (section 4.2.5), and calmodulin works from other groups (section 3.2 and 4.2.5). We also added more discussion about protein-ligand binding in native MS. Please see lines 257-262 and lines 271-273 for detail.

To clarify more, we cleaned up the organization. (1) We included cross references when the same protein was analyzed by different MS techniques and presented results from different techniques in several sections. Taking sidercalin as an example, we now mention how the HDX results motivated the use specific amino acid footprinting to further pinpoint the binding resides (section 4.2.1). We also added the cross referencing to show readers that the HDX discussion is in section 4.1.5. (2) We also rephrased the introductory paragraph (line 50-54) to explain better the organization of this review. (3) We moved the Abeta HDX study from section 3 to HDX-MS section (4.1).

More specific comments:

- l.48-50: I would be good to also discuss contributions by others. Many labs employ covalent labelling, also data on metal binding proteins.

Response: We now include crosslinking works by other labs under the covalent labeling section (section 4.2.5) and added other groups’ covalent labeling footprinting work to cover describe other reagents that were not discussed in the previous text (section 4.2.4).

- l.69ff: HDX should be used as abbreviation for the exchange reaction not the entire process including MS, which is used in a confusing fashion in the following text.

Response: We corrected the terminology throughout the manuscript as per the reviewer’s comment.

- l.94: How about the work from Konermann et al. showing that radical reactions carry on much longer despite scavenger addition?

Response: We now include the discission of secondary radical lifetime in line 95-98. We rephrased the words as “To ensure that the footprinting occurs faster than protein folding or unfolding, the lifetime of primary reactive hydroxyl radicals is limited to ~1 μs depending on scavenger identity and concentration [18, 21], even though some less reactive secondary radicals’ lifetime is milliseconds in solution [22]”.

- authors should check whether sufficient references are provided in section 2, the referencing appears scarce for certain paragraphs.

Response: We now include more references in the section 2.

- 3.1.1. title is a bit confusing is cap-snatching by viral polymerases meant? Only some of these are called L protein. The need for Mn2+ was reported earlier for L proteins, e.g. in Holm et al 2018 JBC. The way it is written now, it suggests that 2020 Wang et al are the first to show this.

Response: We changed the subtitle to “Mn2+ as Co-factor for SFTSV endonuclease”. We added “These observations are consistent with a previous finding that Mn2+ acts as co-factor for other sNSV’ endonucleases [32]” to the end of the paragraph (line 185-186).

- l.174ff: mutation to what? Alanine?

Response: We deleted this sentence for conciseness and added the conclusion about consistence with previous finding. Please see above response for detail.

- 3.2 Abeta metal binding has also been analysed by native MS, e.g. by Young et al, it would be useful to present this alongside.

Response: Thank you for the suggestion. Unfortunately, we could not find this article, but we did include “Metal Ion Binding to the Amyloid β Monomer Studied by Native Top-Down FTICR Mass Spectrometry, Frederik Lermyte, James Everett, Yuko P. Y. Lam, Christopher A. Wootton, Jake Brooks, Mark P. Barrow, Neil D. Telling, Peter J. Sadler, Peter B. O’Connor & Joanna F. Collingwood, Journal of The American Society for Mass Spectrometry volume 30, 2123–2134 (2019).” to cover native top-down MS subject. Please see section 3.3 for detail.

- l.354/355: I agree that HDX and FPOP are superior in localizing binding but doubt that these are superior in determining stoichiometry as they will be subject to similar issues as ITC for stoichiometry determination. Please rephrase slightly.

Response: We rephrased to “Overall, compared with native MS, MS-based titration approaches require more time and thus are not suitable for fast screening of binding metals.” (line 356-357).

- l.364: strictly speaking HDX is not label-free, it just employs the smallest possible label.

Response: We rephrased to “HDX-MS can determine the binding site and conformational changes with high reliability” (line 375-376).

- 4.1.1.: if not mistaken Calmodulin metal binding has also been investigated by native MS, oxidation profile and various other techniques. It would hence be a good example to show the contribution of the various techniques and also showcase work from other groups.

Response: We now include works of calmodulin studied by native MS coupled to IM (section 3.2), oxidation footprinting (section 4.3), and cross-linking (section 4.2.5).

- l.399: only half of the mutation sites are exchanged for alanine.

Response: We have changed from “critical Ca2+ binding residues were substituted with alanine (E111Q/D112N and E103A/D110A).” to “critical Ca2+ binding residues were mutated (E111Q/D112N and E103A/D110A).” (line 414-415).

- l.514: introduce GEE

Response: We added an introduction to GEE (line 596-599). “Glycine ethyl ester (GEE) is one of the most popular choices for the nucleophile, attaching a mass tag of +85 Da to Asp and Glu [105]. It has been employed to study apolipoprotein E, vascular endothelial growth factor, FMO antenna protein, etc. [106-111].”

- scheme 1: the link to the protein is different on left and right side lacking a carbon atom prior to the carbonyl group on the right side.

Response: Thank you for pointing this out. We corrected this.

- l.556-558: also many cross-linking groups have looked into that. Citing some manuscripts here would provide a more balanced review.

Response: We added a cross-linking application for metal ion/protein interaction and cited a comprehensive cross-linking review. Please see section 4.2.5 for detail.

Reviewer 2 Report

The manuscript “Mass Spectrometry-based Structural Proteomics for Metal Ion/Protein Binding Studies" submitted by Yanchun Lin and Michael L. Gross dissects the use of MS-based structural proteomics applied to study metal ion-protein interactions. This is a comprehensive review on this topic, based on recently published research. Metal ion -protein interactions play a large number of important roles in biological processes, ranging from protein structure stabilization to catalytic mechanisms. In addition, several diseases have been associated to metal-binding proteins. In my opinion, this review is of high quality, and it should be published after few minor changes.  

Minor comment

  • Figure 4. Please increase the quality of this Figure. Current version is of low resolution and labels are not readable.
  • Related to my previous comment, please increase the size of the axes in all four plots in Figure 3. The labels in rurrent size are not readable.

Author Response

Reviewer 2

Comments and Suggestions for Authors

The manuscript “Mass Spectrometry-based Structural Proteomics for Metal Ion/Protein Binding Studies" submitted by Yanchun Lin and Michael L. Gross dissects the use of MS-based structural proteomics applied to study metal ion-protein interactions. This is a comprehensive review on this topic, based on recently published research. Metal ion -protein interactions play a large number of important roles in biological processes, ranging from protein structure stabilization to catalytic mechanisms. In addition, several diseases have been associated to metal-binding proteins. In my opinion, this review is of high quality, and it should be published after few minor changes.  

Minor comment

  • Figure 4. Please increase the quality of this Figure. Current version is of low resolution and labels are not readable.

Response: Thank you for the comment. We improved the resolution and labels for Figure 4.

  • Related to my previous comment, please increase the size of the axes in all four plots in Figure 3. The labels in current size are not readable.

Response: We increased the font size of the axes label in 4 HDX plots.

Reviewer 3 Report

Lin and Gross described several MS methods for studying the metal ion/protein binding. The review is accurate, well-written and all topics are covered in a sufficiently clear way, moreover this work offers an insight into the variety of MS approaches and their potential applications.

But as the authors wrote, this review should “enable wider application to characterize other metal-ion and protein interactions” and, in this respect, a brief discussion at the end of each section, especially on the 3, 4 and 5 paragraphs, could help the readers to better understand the broader applications for the MS-specific approaches/methods described in the single section.

Author Response

Reviewer 3

Comments and Suggestions for Authors

Lin and Gross described several MS methods for studying the metal ion/protein binding. The review is accurate, well-written and all topics are covered in a sufficiently clear way, moreover this work offers an insight into the variety of MS approaches and their potential applications.

But as the authors wrote, this review should “enable wider application to characterize other metal-ion and protein interactions” and, in this respect, a brief discussion at the end of each section, especially on the 3, 4 and 5 paragraphs, could help the readers to better understand the broader applications for the MS-specific approaches/methods described in the single section.

Response: Thank you for your insightful comment. To enable wider application and help readers choose the right MS tools, we added discussions to highlight the unique features of each MS tool that can benefit metal ion/protein study in each section (i.e.: line 148-154 for section 3; line 364-373 for section 4; line 718-727 for section 5).